# Genetics and Genomics of Pediatric Pulmonary Arterial Hypertension

**DOI:** 10.3390/genes11101213

**Published:** 2020-10-16

**Authors:** Carrie L. Welch, Wendy K. Chung

**Affiliations:** 1Department of Pediatrics, Columbia University Irving Medical Center, 1150 St. Nicholas Avenue, New York, NY 10032, USA; cbw13@columbia.edu; 2Department of Medicine, Columbia University Irving Medical Center, 622 W 168th St, New York, NY 10032, USA

**Keywords:** genomics, pediatrics, lung disease, pulmonary arterial hypertension

## Abstract

Pulmonary arterial hypertension (PAH) is a rare disease with high mortality despite recent therapeutic advances. The disease is caused by both genetic and environmental factors and likely gene–environment interactions. While PAH can manifest across the lifespan, pediatric-onset disease is particularly challenging because it is frequently associated with a more severe clinical course and comorbidities including lung/heart developmental anomalies. In light of these differences, it is perhaps not surprising that emerging data from genetic studies of pediatric-onset PAH indicate that the genetic basis is different than that of adults. There is a greater genetic burden in children, with rare genetic factors contributing to ~42% of pediatric-onset PAH compared to ~12.5% of adult-onset PAH. De novo variants are frequently associated with PAH in children and contribute to at least 15% of all pediatric cases. The standard of medical care for pediatric PAH patients is based on extrapolations from adult data. However, increased etiologic heterogeneity, poorer prognosis, and increased genetic burden for pediatric-onset PAH calls for a dedicated pediatric research agenda to improve molecular diagnosis and clinical management. A genomics-first approach will improve the understanding of pediatric PAH and how it is related to other rare pediatric genetic disorders.

## 1. Introduction

Pulmonary arterial hypertension (PAH) is a rare disease with an estimated prevalence of 4.8–8.1 cases/million for pediatric-onset [1] and 15–50 cases/million for adult-onset disease [2]. Pathogenic changes in the pulmonary vasculature—including endothelial dysfunction, aberrant cell proliferation, and vasoconstriction—give rise to the clinical consequences of increased pulmonary vascular pressures, increased vascular resistance, heart failure, and premature death [3]. The disease is caused by genetic, epigenetic, and environmental factors, as well as gene–environment interactions wherein genetic contributions to disease risk are modified by environmental exposures. Causal genetic factors for PAH are typically autosomal dominantly inherited for genes such as *BMPR2*, the major gene causing familial forms of PAH (FPAH) [4,5]. Environmental risk factors include hypoxia and exposure to drugs and toxins [4]. Epigenetic factors include active histone mark H3K27ac [6]. Most of our understanding of PAH etiology and treatment is based upon studies in adults [7,8]. However, emerging clinical and genetic data indicate that there are fundamental differences between pediatric- and adult-onset disease.

Pediatric PAH differs from the adult-onset disease in several important aspects, including sex bias, clinical presentation, etiology, and response to therapy [7,8,9]. The frequency of PAH is ~3–4-fold higher in females relative to males for adult-onset disease. However, data from the National Biological Sample and Data Repository for PAH (aka PAH Biobank, Table 1) [10,11] and other studies [12,13] indicate that the frequency of pediatric-onset PAH is similar for females and males, suggesting less dependence on sex-specific factors in children. Children present with increased severity of disease, e.g., elevated mean pulmonary artery pressure (mPAP), decreased cardiac output, and increased pulmonary vascular resistance, compared to adults at diagnosis (Table 1) [10,11]. The clinical manifestations likely reflect the complex etiology of disease in children. While prenatal and early postnatal influences on lung growth and development can contribute to the development of PAH across the lifespan, early developmental influences play a particularly prominent role in pediatric-onset PAH in which patients frequently have complex comorbidities such as congenital heart disease (APAH-CHD), Down syndrome, congenital diaphragmatic hernia (CDH), and other developmental lung diseases, including persistent pulmonary hypertension of the newborn (PPHN) [7,14,15]. Histopathological studies have identified abnormal lung development and lung hypoplasia as common features of PAH, CHD, CDH, and Down syndrome [14,16]. While the mechanisms for impaired lung development are not known, altered expression of angiogenic and anti-angiogenic genes likely contribute [17,18,19,20]. Decreased lung vascular and alveolar growth predispose one to vascular injury during susceptible periods of growth and adaptation. The presentation of pediatric PAH with developmental comorbidities contributes to poor outcomes in these children [7,15]. Few pharmaceutical therapies are approved for use in children due to the lack of safety and efficacy data [8]. However, a retrospective study of pediatric PAH patients suggested that PAH patients with Down syndrome may be less responsive to PAH treatments than non-Down syndrome patients [21]. Clearly, pediatric-focused studies are needed to increase our understanding about the natural history, the pathogenic mechanisms, and the treatment of PAH in children.

## 2. Genetics of Pediatric PAH—Current Knowledge

Emerging data from genetic studies of pediatric-onset PAH indicate that the genetic basis in children is different from that of adults [10,11,13]. There is a greater genetic burden in children, with rare genetic factors contributing to ~42% of pediatric-onset PAH compared to ~12.5% of adult-onset PAH (Figure 1). De novo variants are frequent in children, likely contributing to ~15% of pediatric PAH [10,13]. Among rare inherited variants, variants in *BMPR2* are causal in ~6.5–7% of both pediatric- and adult-onset PAH; most of the cases are FPAH or idiopathic PAH (IPAH), rarely PAH associated with other diseases (APAH) [11,22], and no occurrences in PPHN have been reported to date [23]. Notably, two of the other known causal genes with the highest frequencies of rare deleterious variants among pediatric cases—*TBX4* and *SOX17*—are highly expressed in embryonic tissues and have prominent roles in lung and vasculature development [24,25,26]. The mean age of PAH onset by risk gene is shown for twelve of the genes in Figure 2.

### 2.1. TBX4

Unlike *BMPR2* and other known causal PAH genes, *TBX4* is not expressed in pulmonary arterial endothelial cells or smooth muscle cells. *TBX4* is a transcription factor in the T-box gene family that is co-expressed with *TBX5* throughout the mesenchyme of developing lung and trachea [24]. Lung-specific *Tbx4*/*Tbx5* deficient mice exhibit impaired lung branching and hypoplasia during gestation as well as early postnatal death due to severe respiratory disease [24]. *TBX4* is also expressed in the developing atrium of the heart and the limb buds [27]. In humans, rare but recurrent microdeletions of chromosome 17q23, including *TBX4*, have been observed in children with complex phenotypes including PAH, heart and skeletal defects, and neurodevelopmental delay [28,29,30]. More recently, *TBX4*-specific likely gene-disrupting (LGD) and damaging missense variants have been associated with PAH with or without small patella syndrome (OMIM #147891), most frequently in pediatric cases [11,13,22,31,32]. In two independent cohorts [11,13], rare deleterious variants in *TBX4* showed significant enrichment among pediatric- compared to adult-onset IPAH cases (Columbia University Irving Medical Center, CUIMC, cohort: 10/130 vs. 0/178; PAH Biobank: 12/155 vs. 1/257, respectively). In the PAH Biobank, ten additional *TBX4* variants were identified for other PAH subtypes, including three APAH-CHD cases with heart defects. In a cohort of 256 APAH-CHD cases (144 pediatric- and 112 adult-onset), we identified *TBX4* variants in seven cases with age-of-onset from newborn to 11 years, one associated with alveolar hypoplasia [22]. Together, the data suggest that rare *TBX4* variants contribute to 7.7% of pediatric IPAH and 4.9% of pediatric APAH-CHD cases. Notably, *TBX4* variants have not been observed in CHD alone [33].

Skeletal and other developmental defects are not routinely assessed as part of a PAH diagnosis and, as such, some dual diagnoses may have been missed for PAH cases. To this end, Galambos et al. [34] carried out detailed clinical and histopathologic characterization of 19 pediatric PAH cases with *TBX4* variants: 6 microdeletions, 12 LGD, and 1 missense. Seven infants had evidence of abnormal distal lung development, and there was a high frequency of heart and skeletal developmental anomalies; neurodevelopmental delay was also observed among those patients with microdeletions, likely due to haploinsufficiency of other adjacent genes. Ten newborns presented with PPHN which resolved but recurred later in infancy or childhood [34]. A report from the National French Registry [35] concurred these findings of skeletal, heart, and lung developmental anomalies in PAH cases. Why some patients present with PAH alone, small patella syndrome alone, PAH with small patella syndrome, or PAH with other developmental defects is not understood at this time but may depend on the variant type or the protein location of gene variants, other genetic or epigenetic factors, or other environmental factors affecting the specific transcriptional pathways regulated by TBX4. It *is* clear that genetic diagnosis of a rare deleterious *TBX4* variant or *TBX4*-containing microdeletion in pediatric PAH predicts a more complex developmental phenotype (*TBX4* syndrome [36]). Chest imaging for severe and diffuse features of pulmonary growth arrest, assessment for congenital heart defects, physical examination of hands and feet, and radiological assessment of pelvic areas are recommended. In addition, a *TBX4* diagnosis predicts potential recurrence of PAH following neonatal PPHN suggesting that annual screening by echocardiography may be useful.

### 2.2. SOX17

*SOX17* is a member of the conserved family of SRY-related HMG box transcription factors, originally identified as key regulators of male sex determination but now recognized to have critical roles in embryogenesis [25,26]. *SOX17* is specifically required for endoderm formation and vascular morphogenesis [25,37,38], and germline deletion of *Sox17* results in embryonic lethality by E10.5 [25]. In the developing murine lung, *Sox17* is expressed in mesenchymal progenitor cells and is then restricted to endothelial cells of the pulmonary vasculature [39]. Conditional deletion of *Sox17* in mesenchymal progenitor cells causes abnormal pulmonary vascular morphogenesis, resulting in postnatal cardiopulmonary dysfunction and juvenile death [39]. Endothelial-specific inactivation of *Sox17* in mice leads to impaired arterial specification and embryonic death or, with conditional postnatal inactivation, arterial-venous malformations [37]. Transcriptional activation of *Sox17* via hypoxia-induced factor 1α leads to upregulation of cyclin-E1 and endothelial regeneration in response to lung injury [40]. We identified *SOX17* as a candidate risk gene for PAH using exome sequencing data in a cohort of 256 APAH-CHD patients [22]. Thirteen cases with rare predicted deleterious *SOX17* variants were identified, seven LGD and six missense variants located primarily within the conserved HMG box domain (Figure 2). Fifty-six percent of the overall cohort were pediatric cases, but nine of thirteen cases with rare deleterious variants in *SOX17* were pediatric cases with mean age-of-onset of 14 years. A recurrent frameshift variant, p.(Leu167Trpfs*213), was identified in three APAH-CHD cases with age-of-onset ranging from 7 months to 5 years. We [11,22] and others [41,42] have identified *SOX17* variants in IPAH cases but with lower frequency in adults. Combined data from five cohorts ([11,13,22,41,42] indicate that *SOX17* variants contribute to 7% of all pediatric-onset PAH cases compared to 0.4% of adult-onset cases (Figure 3). Protein modeling indicates that at least three of the APAH-CHD case missense variants localize to the transcription factor DNA binding pocket [22], and missense variants in this region have been shown to impair both direct DNA binding and SOX17/β-catenin nucleoprotein complexes at target gene promoters [43,44]. These data suggest that haploinsufficiency with complete or partial loss of function alleles is the likely mechanism of *SOX17* risk in PAH.

Some variants in SOX17 downstream target genes may be predicted to mimic the consequences of *SOX17* LGD variants or haploinsufficiency. We identified 163 rare predicted deleterious variants in 149 putative SOX17 target genes, most with prominent expression in pulmonary artery endothelial cells and/or developing heart [22]. For the 32 LGD and the 131 missense variants, we observed a moderate but significant enrichment of rare missense variants in cases compared to controls. Approximately one-third of these genes had top quartile gene expression in both pulmonary artery endothelial cells and developing heart. Pathway analysis indicated that the genes have likely roles in developmental biology, small molecule transport/homeostasis, and extracellular matrix interactions (Table 2). While these results are intriguing, they require confirmation in larger cohorts to determine which specific SOX17-regulated genes/pathways contribute to PAH risk.

### 2.3. The Relative Contribution of Other Known PAH Risk Genes

Among the combined CUIMC and PAH Biobank pediatric cohort of 443 non-overlapping cases, rare deleterious variants in other known PAH risk genes altogether account for ~12% of cases (27 IPAH, 26 APAH-CHD, 2 APAH-HHT). Variants in *NOTCH1* account for 2.7% of pediatric PAH cases (2 IPAH, 10 APAH-CHD). *NOTCH1* encodes a transmembrane receptor that facilitates intercellular interactions and signaling with known roles in development and is a known CHD risk gene. Variants in *ABCC8* and *SMAD9* account for 2% and 1% of cases, respectively, including both IPAH and APAH-CHD. Variants in *ACVRL1/ENG* account for 1% of cases, including two APAH-HHT cases. *BMPR1B*, *CAV1*, *GDF2*, *KCNK3* and *KDR/BMP9* were identified in one to four cases each, accounting for <1% for each gene. In addition to autosomal dominant inheritance, recessively inherited *EIF2AK4* variants have been identified in 1–3% of children in European and Chinese cohorts [31,45,46]. In addition, a rare occurrence of recessively inherited *GDF2* variants has been reported for a 3-year-old boy with right heart failure [47]. Autosomal recessive inheritance of other risk variants may cause very early-onset severe PAH, and additional pediatric studies are necessary to evaluate rare recessive genetic etiologies.

### 2.4. De Novo Variants

De novo variants have emerged as an important class of genetic factors underlying rare diseases, especially early-onset severe conditions [15,33,48,49,50], due to strong negative selection decreasing reproductive fitness [51]. We recently assessed the role of rare deleterious de novo variants in pediatric PAH using a cohort of 124 parent-child trios (56% IPAH, 38% APAH-CHD, 6% other PAH) [10]. We observed a 2.5-fold enrichment of de novo variants among all PAH cases compared to the expected rate, almost entirely due to genes that are highly expressed in developing lung or heart (Table 3). Among the PAH cases identified with de novo variants, 54% were IPAH, 32% were APAH-CHD, and 14% other PAH; at least 20% of the de novo variant carriers had additional diagnoses of other congenital anomalies. De novo variants were identified in three known PAH risk genes (four variants in *TBX4*, two in *BMPR2*, one in *ACVRL1*) and 23 additional genes with high expression in developing lung and/or heart but little to no previous association with PAH. Based on the enrichment rate, we estimate that ~18 of the identified variants are likely to be implicated in pediatric PAH. The identified genes fit a general pattern for developmental disorders—genes intolerant to LGD variants (pLI >0.5 for 40% of the PAH genes) and with known functions important for coordinated organogenesis, including transcription factors, RNA binding proteins, protein kinases, and chromatin modification. Three of the genes are known CHD risk genes (*NOTCH1*, *PTPN11*, and *RAF1*). *NOTCH1* is the most commonly associated gene for the congenital heart defect of tetralogy of Fallot, [52] and the *NOTCH1* de novo variant carrier had a diagnosis of APAH-CHD with tetralogy of Fallot. Rare variants in *PTPN11* and *RAF1* are causal for Noonan syndrome, which has a high frequency of congenital heart defects. The de novo variants identified in both of these genes are known causal Noonan syndrome variants [53], and three cases of fatal pediatric PAH with Noonan syndrome have been previously reported [54,55]. We previously reported rare inherited variants in *NOTCH1* (*n* = 5), *PTPN11* (*n* = 1), and *RAF1* (*n* = 2) carried by APAH-CHD cases [22]. Aside from known PAH and CHD genes, at least eight of the other genes with identified de novo variants have known or plausible roles in lung/vascular development (Table 4). For example, *AMOT* (angiomotin) encodes an angiostatin-binding protein involved in embryonic endothelial cell migration and tube formation as well as endothelial cell tight junctions and angiogenesis [56,57,58]. *HSPA4* (heat shock protein A4) encodes a chaperone that, together with HSPA4L, functions in embryonic lung maturation and dual deletion of *Hspa4*/*Hspa4l*, which results in intrauterine pulmonary hypoplasia and early neonatal death [59]. *KEAP1* (Kelch-like ECH associated protein 1) regulates oxidative stress and apoptosis through interactions with NRF2 in murine vascular cells [60], and endothelial-specific deletion of *NRF2* reduces endothelial sprouting in vivo [61] and increases susceptibility to bronchopulmonary dysplasia and other respiratory diseases [62]. An NRF2 activator is currently being investigated in a phase 2 clinical trial for PAH (ichgcp.net/clinical-trials-registry/NCT02036970). Finally, one third of all of the de novo variants identified in the trio analysis are in causal genes for developmental syndromes, consistent with the enrichment of developmental phenotypes among the variant carriers [10]. The genes identified in this study require replication in a larger pediatric cohort. In addition, genes with rare variants can be entered into GeneMatcher to identify other cases with rare variants in the same gene and compare genotypes and phenotypes. Due to the low background rate of rare deleterious de novo variants [63], the statistical evidence for a candidate risk gene is effectively equivalent to multiplicity. That is, genes with ≥2 rare deleterious de novo variants are unlikely to be mutated by chance and should be considered candidate risk genes. The genes and the variants identified in the pediatric trio analysis have not been observed in adult-onset cases and likely will be specific to pediatric PAH. Thus, it is imperative that larger pediatric-focused PAH cohorts are studied to advance our knowledge of the causal genes specific to pediatric-onset PAH.

### 2.5. Genetic Ancestry

Most of the large genetic studies conducted to date have utilized cohorts of predominantly European ancestry. However, the role of specific genes in PAH may be heterogeneous across genetic ancestries, and the results of these studies may not be generalizable to all other populations. For example, the frequency of *ACVRL1* and *ENG* variants combined is ~1% among pediatric IPAH cases of European ancestry [11,13], but the frequency of *ACVRL1* alone may be closer to 13% among Asian children [64]. *GDF2/BMP9* was recently identified as a novel PAH risk gene with genome-wide significance in both European [41] and Asian [65] cohorts with replication in the PAH Biobank cohort [11]. Similar to other PAH risk genes, the mode of inheritance was autosomal dominant. The frequency of *GDF2/BMP9* variants among children was 2.1% (2/94 cases) in the PAH Biobank and 5.2% (3/57 cases) in the Asian cohort, suggesting that *GDF2/BMP9* variants might be a more frequent cause of PAH among Asian children. Further study is required to determine whether this difference is a true genetic ancestry effect or random variation due to relatively small sample size or differences in bioinformatic pipelines. A PAH case study of a five-year-old boy of Hispanic ancestry identified a homozygous *GDF2*/*BMP9* LGD variant, NM_016204.1:c.76C > T; p.(Gln26Ter) [47]. The unaffected parents were heterozygous for the variant. Interestingly, the gnomAD population database (gnomADv2.1.1, *n* = 141,456 samples) [66] contains only two heterozygous counts of this allele, both of Latino ancestry, suggesting that this might be an ancestry-specific allele. Clearly, larger studies of children with greater diversity are needed to define population-specific risk gene allele frequencies as well as ancestral-specific genetic factors.

### 2.6. The Role of Other “Omics” in PAH

In addition to DNA sequencing to identify genetic etiologies of PAH, other “omics”, including RNA sequencing, metabolomics, and proteomics, can provide valuable predictions of who is at risk for disease, define endophenotypes, and guide effective therapies [67,68]. For example, West and colleagues performed RNA sequencing of blood lymphocytes derived from *BMPR2* variant carriers with and without PAH to identify transcriptional patterns relevant to disease penetrance [69]. More recently, *FHIT* was identified as a potentially clinically relevant BMPR2 modifier gene through an siRNA screen of BMPR2 signaling regulatory genes combined with publicly available PAH RNA expression data. Subsequently, the authors showed that pharmaceutical upregulation of FHIT prevented and reversed experimental pulmonary hypertension in a rat model [70]. Rhodes and colleagues utilized metabolomics to identify circulating metabolites that distinguish PAH cases from healthy controls, to predict outcomes among PAH cases, and to monitor metabolite levels over time to determine whether correction could affect outcomes [71]. Stearman et al. combined gene expression data with pathway analyses to identify a transcriptional framework for PAH-affected lungs [72]. Similarly, Hemnes and colleagues used transcriptomics to identify RNA expression patterns predictive of vasodilator responsiveness among PAH patients [73]. These studies highlight the promise of other omics in predictions of PAH risk, diagnosis, classification, drug responsiveness, and prognosis. However, such studies have not been conducted in children. Detailed omic phenotyping requires biologic sampling, which can be difficult in pediatric patients, especially for the very young or those with complex medical conditions. We propose a pilot genomics-first approach followed by detailed phenotyping of patients grouped by genetic diagnosis to enrich the biologic sampling and assess utility before performing larger studies across all pediatric PAH patients.

## 3. A Genomics First Approach towards Better Understanding of Pediatric PAH

Identification of molecular subtypes of PAH has been proposed as a means to improve risk stratification, treatment, and outcomes. Obtaining a genetic diagnosis in children requires more extensive genetic testing than in adults (Figure 4). If testing for a panel of genes known to be associated with PAH is not diagnostic, children should be evaluated genome-wide for rare de novo and inherited variants with trio (parental and child) exome sequencing/chromosome microarray or genome sequencing. With knowledge of the causal gene, natural history, penetrance, and response to treatment can be refined for that specific genetic subtype of PAH to allow for more precise care for each genetically defined group. Individuals across genes in the same biological pathway can then be compared to assess similarities and differences.

Collaboration across national and international clinical PH sites will be necessary to yield sufficient sample sizes due to the extremely small number of pediatric PAH patients at single PH sites, heterogeneity of risk genes for PAH, and need for ancestral diversity. PPHNet is an example of a pediatric-specific PAH consortium with ongoing recruitment across 13 North American clinical sites [74,75,76,77,78]. PVDomics [79], a US multicenter study launched in 2014, and PAH-ICON (pahicon.com), a new international effort, represent additional large-scale PAH cohorts. PAH was defined as mPAP ≥25 mmHg but was recently updated to include mPAP 20–25 mmHg in both children and adults [8]. Due to variability in pulmonary hemodynamics during the post-natal transition period, pediatric PAH is defined by elevated mPAP after 3 months of age in combination with pulmonary vascular resistance as indexed to body surface area, PVRI ≥3 Woods units/m^2^ [8]. Clinical classification of PAH subtypes aims to improve clinical management and enhance research efforts and is typically based on the World Symposium on Pulmonary Hypertension (WSPH) system, updated during the 2018 Nice session [80]. In children, use of a pediatric-specific classification system developed in Panama by the Pulmonary Vascular Research Institute (PVRI) Pediatric Task Force [81] provides more definitive classification of developmental and complex phenotypes.

Biological trios composed of two unaffected parents and an affected child are preferred over singleton cases for pediatric studies of PAH in order to identify both inherited and de novo variants causal for disease. However, as the number of ethnically matched genomic data in public databases increases, the need for trios will decrease. DNA can be reliably obtained from small samples of blood or saliva. We have developed methods in a large national autism study to collect saliva from pediatric patients and their parents in their homes, with instructional videos to support clinical sites with remote biospecimen collection [82,83]. Biological samples can be shipped to a central biorepository for DNA extraction and then processed and sequenced using a single genetic platform. Since annotation tools for predicting relative pathogenicity of noncoding variants are still under development, and because the incremental yield of structural variants identified from genome sequencing is low, there is currently little added value in analysis of genome sequencing compared to exome sequencing data. Following extensive quality control, filtering, and annotation, sequencing data are screened for rare deleterious variants in known candidate genes or undergo trio analysis for de novo variants or association analysis for inherited variants [10,11]. Candidate genes identified by these methods are further assessed by mapping the locations of variants to protein structures, assessing expression in PAH-relevant tissues and cell types, assessing variant function in vitro, and assessing the impact of pathogenic variants in vivo in model organisms. Human mutations can be introduced into cells and PAH-sensitized mice using CRISPR technology. Phenotypic “rescue” by exogenous delivery of normal proteins can add evidence to support causation.

Once genetic subtypes are defined, demographic data, clinical phenotypes, and imaging data (pulmonary vascular angiography, chest X-ray or CT scans, chest/cardiac MRI, and lung biopsies) can be compared among cases with variants in the same gene/pathway. Relevant cases can be recalled or remotely interviewed for targeted clinical assessments to determine if there are similarities among cases with rare variants in the same gene/pathway.

To increase rigor and assess the full phenotypic spectrum of the new genetic subtypes, additional cases can be identified using GeneMatcher, clinical diagnostic laboratories, and large sequencing centers. Longitudinal phenotypes of the genetic subtypes can be assessed retrospectively and prospectively, including death/transplant, response to medication, other medical diagnoses, and changes in lung function. For inherited variants, cascade genetic testing of family members with clinical evaluation of “unaffected” individuals who carry the relevant genetic variant can inform penetrance by age and sex. To support families with genetic diagnoses, “virtual” family meetings can be organized to update families on new findings related to their conditions and build communities for each of the rare endophenotypes.

The value of a genetic diagnosis to families is three-fold: (**1**) identification of other associated features, (**2**) identification of family members at risk for developing PAH, and (**3**) clarification of reproductive risks and provide family planning options. Biallelic mutations in *EIF2AK4* are diagnostic for pulmonary veno-occlusive disease/pulmonary capillary hemangiomatosis [84], which can be difficult to diagnose clinically without a lung biopsy, and patients can be listed for transplant earlier in the course of disease, which may improve outcomes. As mentioned, *TBX4* variant carriers should be assessed for associated developmental (lung, heart) and orthopedic (hips, knees, feet) issues. *ENG/ACVRL1* variant carriers are prone to arteriovenous malformations in brain, intestine, lung, and liver [85]; these patients require periodic MRI surveillance. After making a genetic diagnosis in a PAH patient, additional family members can be screened for the family variant to identify those at risk who may benefit from annual surveillance and early diagnosis/treatment. Furthermore, diagnosed young adults can make informed decisions regarding family planning.

## 4. Conclusions

Pediatric PAH differs from adult-onset PAH in many important aspects, including clinical presentation, etiology, genetic burden, and specific genes involved. In many young children, PAH is a developmental disease with a complex phenotype. *TBX4* and *SOX17* are examples of developmental genes in which rare deleterious variants occur much more frequently in pediatric- compared to adult-onset PAH. De novo variants likely contribute to at least 15% of pediatric-onset PAH, but the specific genes require confirmation in larger pediatric cohorts. Many genes with de novo variants likely contribute to developmental phenotypes and complex medical conditions. A genomics-first approach to pediatric PAH starts with a genetic diagnosis followed by phenotypic characterization of cases with variants in the same genes/pathways. Large, diverse pediatric populations are needed to confirm the candidate genes identified thus far, identify new genes, characterize each rare endophenotype and natural history, and assess the efficacy of therapies to inform more precise clinical management. In addition, questions related to which children are at risk for developing PAH—especially children with CHD, CDH, bronchopulmonary dysplasia, and Down syndrome—may be answered. The yield of genetic diagnoses in pediatric-onset PAH cohorts is significantly greater than the yield in adult-onset cohorts. However, identification of genes, pathways, and networks in children could provide novel targets for therapy not only for children but for all patients with and at high risk for PAH.

## Figures and Tables

**Figure 1 genes-11-01213-f001:**
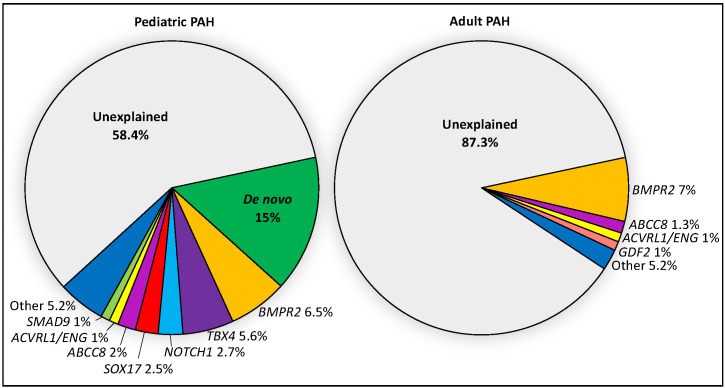
Relative contributions of *de novo* mutations and 18 PAH risk genes in a cohort of 443 pediatric and 2628 adult cases from CUIMC and the PAH Biobank. Risk genes include *BMPR2*, *ABCC8*, *ACVRL1*, *ATP13A3*, *BMPR1B*, *CAV1*, *EIF2AK4*, *ENG*, *GDF2*, *KCNA5*, *KCNK3*, *KDR*, *NOTCH1*, *SMAD1*, *SMAD4*, *SMAD9*, and *TBX4*. PAH cases include IPAH, APAH, FPAH and other rarer cases.

**Figure 2 genes-11-01213-f002:**
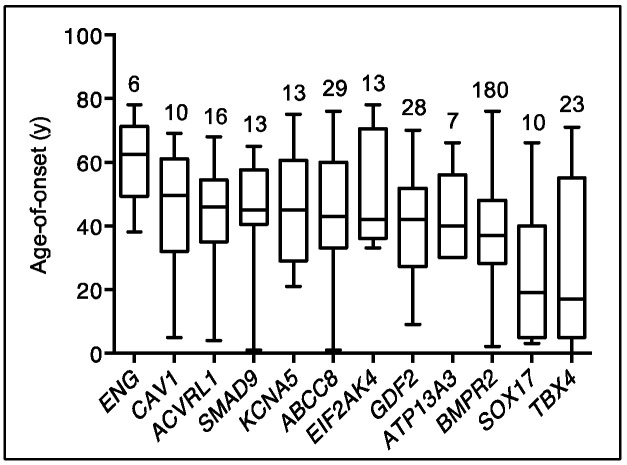
Age-of-disease onset for all PAH Biobank cases with rare deleterious variants in known PAH risk genes. Box plots showing median, interquartile range and min/max values for age-of-disease onset (i.e., age at diagnostic right heart catheterization). The number of cases carrying variants for each gene is given above each box plot. Genes represented by less than four cases are not shown.

**Figure 3 genes-11-01213-f003:**
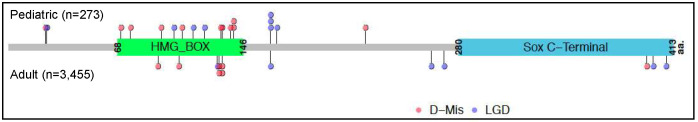
Locations of SOX17 likely gene disrupting (LGD) and rare predicted deleterious missense (D-Mis) variants carried by PAH cases from five cohorts from the US, UK and Japan. Variants carried by pediatric cases (*n* = 19) are shown above the protein schematic and variants carried by adult cases (*n* = 13) below the schematic. The combined datasets include 273 pediatric and 3455 adult cases [11,13,22,41,42].

**Figure 4 genes-11-01213-f004:**
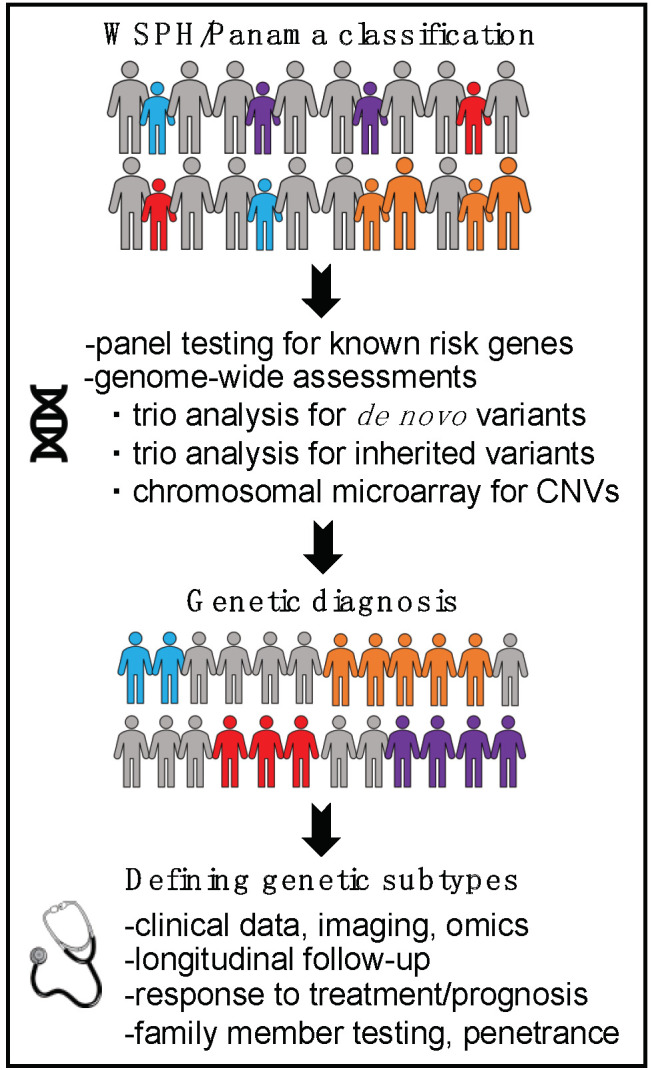
Genomic approach to improve understanding of pediatric PAH.

**Table 1 genes-11-01213-t001:** Clinical characteristics and hemodynamic parameters of child- vs. adult-onset pulmonary arterial hypertension (PAH) cases at diagnosis. Data are from the PAH Biobank (*n* = 2572). Child-onset, <18 years of age at diagnosis. Mean ± SD.

Group (n)	Age at dx (y)	F:M Ratio	mPAP (mm Hg)	mPCWP (mm Hg)	CO Fisk (L/min)	PVR (Woods Units)	Common Comorbidities
Child (226)	7.7 ± 5.4	1.65:1	55.1 ± 18.6	9.0 ± 3.0	3.2 ± 1.6	18.1 ± 11.7	CHD, CDH, DS, lung growth/development
Adult (2345)	51.6 ± 14.7	4.02:1	49.6 ± 13.9	10.2 ± 4.2	4.6 ± 1.7	10.0 ± 5.9	HTN, hypothyroidism, other pulmonary & metabolic diseases
*p*-value	<0.0001 *	<0.0001 **	<0.0001 *	<0.0001 *	<0.0001 *	<0.0001 *	

Abbreviations: dx, diagnosis; F:M, female:male; mPAP, mean pulmonary artery pressure; mCWP, mean pulmonary capillary wedge pressure; CO, cardiac output; PVR, pulmonary vascular resistance; CHD, congenital heart disease; CDH, congenital diaphragmatic hernia; DS, Down syndrome; HTN, systemic hypertension. * Student’s t-test, 2-tailed. ** Fisher exact test.

**Table 2 genes-11-01213-t002:** Biological pathway analysis of SOX17 target genes harboring PAH-CHD patient rare deleterious variants. Data obtained using Reactome 2016. Pathways with false discovery rate (FDR)-adjusted *p*-value ≤ 0.05 are listed.

Term	Reactome ID	# Genes in Overlap	*p*-Value	Adjusted *p*-Value	Genes
Developmental biology	R-HAS-1266738	16/786	6.8 × 10^−5^	0.03	*KLB*, *ROBO2*, *LAMA1*, *EGF*, *ANK3*, *LAMC*, *SLC2A4*, *MED6*, *SPRED2*, *MEIS1*, *NRFA2*, *PCMC4*, *NF1*, *EP300*, *TCF4*, *EPHB4*
Transmembrane transport of small molecules	R-HAS-382561	13/594	1.7 ×10^−4^	0.03	*RYR2*, *ABCC4*, *ABCC1*, *SLC1A3*, *SL#3A4*, *SLC8A1*, *CLCN5*, *SLCA9*, *ATPB7*, *ASPH*, *WNK1*, *NUP35*, *EMB*
Non-integrin membrane extracellular matrix interactions	R-HAS-3000171	4/42	1.7 × 10^−4^	0.03	*LAMA1*, *LAMA4*, *LAMC1*, *THBS1*
Ion homeostasis	R-HAS-5578775	4/51	1.7 × 10^−4^	0.03	*RYR2*, *ASPH*, *TPR3*, *SLC8A1*

**Table 3 genes-11-01213-t003:** Burden of de novo variants in 5756 genes highly expressed in developing lung (murine E16.5 lung stromal cells) and/or developing heart (murine E14.5 heart) in pediatric-onset PAH (*n* = 124 child/parent trios).

Variant Type *	Observed in Trios (*n* = 124)	Expected by Chance	Enrichment	*p*-Value	Estimated # of True Risk Variants
SYN	18	14.0	1.3	0.28	---
LGD	11	4.7	2.4	0.06	---
MIS	40	31.7	1.3	0.15	---
D-MIS	19	7.2	2.6	2.0 × 10^−4^	12
LGD + D-MIS	30	11.8	2.5	7.0 × 10^−6^	18

* SYN, synonymous; LGD, likely gene disrupting; MIS, missense; D-MIS, predicted deleterious missense based on REVEL score > 0.5.

**Table 4 genes-11-01213-t004:** Novel genes with rare deleterious de novo variants in pediatric-onset PAH (*n* = 124 trios).

Gene Symbol	Variant Type	Protein Change	REVEL Score	CADD Score	Allele Frequency (gnomAD)	E16.5 Lung Expression Rank	E14.5 Heart Expression Rank	Variant Carrier PAH Subtype
*AMOT*	LGD	p.(Leu320Cysfs*55)	.	31	.	68	95	IPAH
*CSNK2A2*	D-MIS	p.(His184Leu)	0.50	25	.	55	77	IPAH
*HNRNPF*	LGD	p.(Tyr210Leufs*14)	.	29	.	85	98	PPHN, PAH
*HSPA4*	D-MIS	p.(pro684Arg)	0.62	30	4.1 × 10^−6^	43	96	PAH-CHD
*KDM3B*	D-MIS	p.(Pro1100Ser)	0.66	29	.	89	87	IPAH
*KEAP1*	LGD	p.(Tyr584*)	.	35	.	79	82	IPAH with dev delay
*MECOM*	D-MIS	p.(Phe762Ser)	0.76	32	.	82	60	IPAH
*ZMYM2*	LGD	p.(Arg540*)	.	36	.	93	77	IPAH with skeletal anomalies

*AMOT*, angiomotin; *CSKN2A2*, casein kinase II, α 2; *HNRNPF*, heterogeneous nuclear ribonucleoprotein F; *HSPA4*, heat shock protein A (HSP70), member 4; *KDM3B*, lysine demethylase 3B; *KEAP1*, Kelch-like ECH-associated protein 1; *MECOM*, MDS1 and EVI1 complex locus; *ZMYM2*, zinc finger protein 620. LGD, likely gene disrupting; MIS, missense; D-MIS, predicted deleterious missense based on REVEL score > 0.5. Allele frequency “.” absent from gnomAD.

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
