# Peer review of "Genetics and Genomics of Pediatric Pulmonary Arterial Hypertension"

_genes, 2020, doi:10.3390/genes11101213_

Round 1

Reviewer 1 Report

Genetics and Genomics of Pediatric Pulmonary Arterial Hypertension

Carrie L. Welch and Wendy K. Chung

This is a review on pediatric pulmonary hypertension and the genetic underpinnings written by Dr. Chung, a well-regarded physician scientist in genomic and precision medicine. The area of genetics and PAH is of specific interest as increasing genetic etiologies are identified and the overall prognosis remains poor. New genetic etiologies are being identified.

Major comments:

  1. This review begins with the varying etiologies underlying PAH but really is a genetic review of idiopathic PAH rather than structural etiologies (congenital heart disease or congenital diaphragmatic hernia for instance). It is unclear in the shift from all PAH to IPAH what percentage of PAH is IPAH and what overlap exists between genetic and structural IPAH.
  2. It is unclear in the discussion if the authors are advocating for genetic workup in all patients with PAH even if they have a presumed underlying diagnosis – be that structural CDH/CHD or due to a presumed developmental diagnosis (extreme prematurity).
  3. Prematurity which is one of the most common causes of PAH is not discussed.
  4. There is limited discussion of how novel variants thought to underly IPAH are confirmed either through bioinformatic or experimental methods.
  5. There is a focus on SOX17 and TBX5 as etiologies of IPAH, however is unclear why SOX17 which is less common than SMAD9, KCNK3 and ACVRL1 mutations is a focus when the others are not discussed at depth (other than the authors publish on SOX17).
  6. This review would benefit from a discussion of the benefits of genetic diagnosis for patients. For instance with some genetic diagnoses there are other associated sequelae that requiring monitoring (eg, ACVRL1 patients are prone to cerebral AVMs requiring periodic MRI surveillance), and whether genetic diagnosis results in changes in clinical management such as selection of PAH agents.

Minor comments:

None

Reviewer 2 Report

It is a great article, with the focus on the pediatric onset of PAH, which shows how different is the genetic background between child and adult PAH. It has been well written and has special importance to pediatric cardiologist, geneticist, and all professional involved in the diagnosis and management of pediatric patients with PAH.

I have some minor specific questions:

  • How the authors explain the overlapping between PAH and THH in pediatric patients with variants in ACVLR1/ENG?.
  • About 64% of pediatric patients with IPAH do not present any pathogenic variant. How much of this percentage can be due to another genetic variants in not related genes or due to other causes?. In addition, what they think would be the increased in yield by performing WGS instead of WES?.
  • How the authors explain why some cases with pathogenic variants only develop small patella syndrome (SPS) while other manifest not only SPS but also PAH? They have any idea about this?. It is recommended to include a guideline for skeletal follow up in PAH patients having a pathogenic variants in TBX4?.
  • GDF2 can also be considered to have AR inheritance in rare severe pediatric cases as some cases were described?
  • About the De Novo mutations representing around 15% in pediatric PAH patients, how the authors proposed to screen and report these cases in daily clinical routine?

And some comments:

  • The authors do not described neither the role nor the pathway from other genes also relevant in pediatric PAH such as ion channel genes (i.e. KCNK3, KCNA5, ABCC8).
  • The authors do not discuss about the possible role of EIF2AK4 in pediatric cases of severe Pulmonary Venooclusive Disease (PVOD). They will consider this gene only in adult cases? Or what proportion of patients with early onset of PVOD might be due to mutation in this gene according authors experience?.
  • There are relationship between the genetic defect and the phenotype in pediatric PAH that can be helpful for diagnosis and to guide the genetic screening?
